# Identification of Antibacterial Components in the Methanol-Phase Extract from Edible Herbaceous Plant *Rumex madaio* Makino and Their Antibacterial Action Modes

**DOI:** 10.3390/molecules27030660

**Published:** 2022-01-20

**Authors:** Yue Liu, Lianzhi Yang, Pingping Liu, Yinzhe Jin, Si Qin, Lanming Chen

**Affiliations:** 1Key Laboratory of Quality and Safety Risk Assessment for Aquatic Products on Storage and Preservation (Shanghai), Ministry of Agriculture and Rural Affairs of the People’s Republic of China, College of Food Science and Technology, Shanghai Ocean University, Shanghai 201306, China; M190300758@st.shou.edu.cn (Y.L.); D210300069@st.shou.edu.cn (L.Y.); ppliu@shou.edu.cn (P.L.); yzjin@shou.edu.cn (Y.J.); 2Key Laboratory for Food Science and Biotechnology of Hunan Province, College of Food Science and Technology, Hunan Agricultural University, Changsha 410128, China

**Keywords:** *Rumex madaio* Makino, antibacterial component, antibacterial mode, pathogenic bacteria, transcriptome, edible plant

## Abstract

Outbreaks and prevalence of infectious diseases worldwide are some of the major contributors to morbidity and morbidity in humans. Pharmacophageous plants are the best source for searching antibacterial compounds with low toxicity to humans. In this study, we identified, for the first time, antibacterial components and action modes of methanol-phase extract from such one edible herbaceous plant *Rumex madaio* Makino. The bacteriostatic rate of the extract was 75% against 23 species of common pathogenic bacteria. The extract was further purified using the preparative high-performance liquid chromatography (Prep-HPLC) technique, and five separated componential complexes (CC) were obtained. Among these, the CC 1 significantly increased cell surface hydrophobicity and membrane permeability and decreased membrane fluidity, which damaged cell structure integrity of Gram-positive and -negative pathogens tested. A total of 58 different compounds in the extract were identified using ultra-HPLC and mass spectrometry (UHPLC-MS) techniques. Comparative transcriptomic analyses revealed a number of differentially expressed genes and various changed metabolic pathways mediated by the CC1 action, such as down-regulated carbohydrate transport and/or utilization and energy metabolism in four pathogenic strains tested. Overall, the results in this study demonstrated that the CC1 from *R. madaio* Makino are promising candidates for antibacterial medicine and human health care products.

## 1. Introduction

China is one of the richest countries in biodiversity, with very high levels of plant endemism [1]. Pharmacopoeia of the Peoples’ Republic of China (2020 Edition) contains 2711 species of Chinese herbal plants, which constitute a gold mine for exploiting medicine candidates and health care products [2]. For instance, *R. madaio* Makino is an edible, perennial and herbaceous plant that belongs to the *Dicotyledoneae* class, *Polygonaceae* family, and *Rumex* genus. According to the National Compilation of Chinese Herbal Medicine (1996 Edition), leaf and root tissues of *R. madaio* Makino can be used as medicine such as clearing heat and detoxification, removing blood stasis, and defecating and killing insects. Nevertheless, current studies on the antibacterial activity of *R. madaio* Makino are rare.

In this study, antibacterial components and action modes of methanol-phase extract from *R. madaio* Makino were for the first time identified. The objectives of this study were: (1) to extract bioactive substances from *R. madaio* Makino using the methanol and chloroform extraction (MCE) method, and determine their inhibition activity against 23 species of pathogenic bacteria; (2) to purify the methanol-phase extract from *R. madaio* Makino by preparation high-performance liquid chromatography (Prep-HPLC) analysis, and identify bioactive compounds in componential complex 1 (CC 1) using an ultra-HPLC and mass spectrometry (UHPLC-MS) technique; (3) to determine cell surface hydrophobicity, cell membrane permeability, fluidity, and the damage of four representative pathogenic strains treated with the CC 1; (4) to decipher possible molecular mechanisms underlying antibacterial activity by comparative transcriptomic analysis. The results of this study meet the increasing need for novel antibacterial agent candidates against common pathogenic bacteria.

## 2. Results and Discussion

### 2.1. Antibacterial Activity of Crude Extracts from R. madaio Makino

Antibacterial substances in fresh leaf and stem tissues of *R. madaio* Makino were extracted using the MCE method. The results showed that the water loss rate of the plant material was 93.32%, and extraction rates of the methanol phase and chloroform phase were 32.10% and 29.60%, respectively. Antibacterial activity of the crude extracts against 23 species of pathogenic bacteria was determined, most of which are common foodborne pathogens, and the results are presented in Table 1. The chloroform-phase crude extract from *R. madaio* Makino showed a bacteriostatic rate of 39%, inhibiting 2 species of Gram-positive and 11 species of Gram-negative pathogens (Table 1, Figure 1). Remarkably, the methanol-phase crude extract from *R. madaio* Makino inhibited the growth of 33 bacteria strains tested with a bacteriostatic rate of 75%, including 2 species of Gram-positive and 18 species of Gram-negative pathogens (Table 1). Based on the higher bacteriostatic rate (75%), the methanol-phase crude extract from *R. madaio* Makino was chosen for further analysis in this study.

### 2.2. Purification of the Methanol-Phase Crude Extract from R. madaio Makino

Large amounts of the methanol-phase crude extract from *R. madaio* Makino were further purified by the Prep-HPLC analysis. As shown in Figure 2, five obviously separated peaks (designated as componential complex, CCs 1 to 5) were observed by scanning at OD_280 nm_ for 15 min.

These five single peaks were individually collected for antibacterial activity analysis. The results revealed that the CC 1 had strong inhibitory effects on *Vibrio parahaemolyticus* ATCC17802, *Vibrio alginolyticus* ATCC17749, *Bacillus cereus* A1-1, and *V. parahaemolyticus* B4-10. Moreover, the growth of the other four strains was also depressed, including *V. parahaemolyticus* ATCC33847, *V. parahaemolyticus* B3-13, *V. parahaemolyticus* B5-29, and *Staphylococcus aureus* ATCC6538 (Table 2). Among these, *V. alginolyticus* is an opportunistic pathogenic bacterium that can infect a broad range of marine host animals, including fish, crab and pearl oysters, and can also infect the human ear, soft tissue and wounded sites [3,4], while *V. parahaemolyticus* is a leading seafood-borne pathogen worldwide and can cause acute gastroenteritis and septicemia in humans [5]. *B. cereus* is a Gram-positive bacterium for food poisoning. This bacterium has been incriminated in clinical conditions such as anthrax-like progressive pneumonia, fulminant sepsis, and devastating central nervous system infections, particularly in immunosuppressed individuals, intravenous drug abusers, and neonates [6].

Conversely, the other four peaks (CCs 2 to 4) showed weak or no antibacterial activity, indicating that bioactive compounds in the methanol-phase extract from *R. madaio* Makino existed in the CC 1.

MIC values of the CC 1 were also determined, which was 64 μg/mL against *V. alginolyticus* ATCC17749 and *V. parahaemolyticus* ATCC17802; 128 μg/mL against *B. cereus* A1-1; and 256 μg/mL against *V. parahaemolyticus* B4-10.

### 2.3. Changed Bacterial Cell Surface Structure by the CC 1 Extract

To decipher possible mechanisms underlying bacteriostatic activity of the CC 1, the cell structure of the four highly inhibited strains were observed by the transmission electron microscope (TEM) analysis. As shown in Figure 3, in remarkable contrast to control groups whose cell surface structure was intact, showing rod cells, a flat surface, and a clear structure, bacterial cells in the treatment groups showed different degrees of contraction and rupture, some of which were deformed with obvious depressions, folds or cavities on the surface. For example, for the Gram-positive *B. cereus* A1-1, the 2 h treatment by the CC 1 resulted in the bacterial cell surface shrinking seriously, the flagella breaking, and some contents leaking. After being treated for 4 h, cell surface shrinkage was intensified, and more cells were ruptured. After being treated for 6 h, the cell structure was seriously damaged, a large number of contents exuded, and only a few cells still maintained rod shape (Figure 3A). For the Gram-negative *V. parahaemolyticus* ATCC17802, after being treated with the CC 1 for 2 h, its cell surface shrunk slightly, and pili structure was still visible. However, after being treated for 4 h, the cell surface shrinkage increased and the cell membrane folded. *V. parahaemolyticus* ATCC17802 cells were destroyed, seriously shrunk and deformed after being treated for 6 h (Figure 3C). These results indicated that the CC 1 from *R. madaio* Makino damaged the cell surface structure of the Gram-negative and Gram-positive pathogens.

### 2.4. Changed Bacterial Cell Surface Hydrophobicity, Cell Membrane Fluidity, Permeability, and Damage by the CC 1 from R. madaio Makino

Cell surface hydrophobicity plays an important role in the adhesion to abiotic and biological surfaces and infiltration of host tissue [7]. In this study, bacterial cell surface hydrophobicity of all four experimental groups was significantly increased (*p* < 0.05) when compared with the control groups (Figure 4A). The effect was highly enhanced with the increase in treatment time. For example, cell surface hydrophobicity was significantly increased in *V. parahaemolyticus* ATCC17802 (1.47-fold), *V. parahaemolyticus* B4-10 (1.62-fold) and *B. cereus* A1-1 (1.42-fold) after being treated with the CC1 for 2 h (*p* < 0.05), whereas a similar change was observed in the treatment group of *V. alginolyticus* ATCC17749 (1.48-fold) after being treated for 4 h. Moreover, the highest increase in cell surface hydrophobicity was observed in *B. cereus* A1-1 (3.75-fold) after being treated with the CC1 for 6 h (Figure 4A).

Membrane fluidity is also a key parameter of the bacterial cell membrane that undergoes quick adaptation in response to environmental challenges [8]. It has recently been regarded as an important factor in the antibacterial mechanism of membrane-targeting antibiotics [9]. In this study, compared with the control groups, there was no significant difference in cell membrane fluidity of *V. parahaemolyticus* ATCC17802 and B4-10, as well as *V. alginolyticus* ATCC17749 after being treated with the CC 1 for 2 h (*p* > 0.05). However, a significant decrease in membrane fluidity of these three strains was observed after the treatment for 4 h. Additionally, cell membrane fluidity significantly declined in *B. cereus* A1-1 (1.20-fold) treated with the CC 1 for 2 h, and sharply lost for 6 h (8.11-fold) (Figure 4B). The change of membrane lipid composition likely contributed to the observed membrane fluidity change to resist the lipid disorder effect by therapeutic agents [10].

The o-nitrophenyl-β-d-galactopyranoside (o-nitrophenyl)-β-d-galactopyranoside (ONPG) was used as a probe to monitor the inner cell membrane permeability of the four bacterial strains, and the results were illustrated in Figure 5. Different influence of the CC 1 from *R. madaio* Makino on inner cell membrane permeability was observed among the four treatment groups. For example, *V. alginolyticus* ATCC17749 did not change significantly in the inner cell membrane permeability after the treatment for 2 h (*p* > 0.05), whereas a significant increase was observed after being treated for 4 h (1.15-fold) and 6 h (1.18-fold), respectively (*p* < 0.05) (Figure 5).

*N*-Phenyl-1-naphthylamine (NPN) was used as a probe to monitor the bacterial outer membrane permeability. As shown in Figure 6, the outer membrane permeability in the four experimental groups were all highly increased after the treatment with the CC 1 for 2 h (*p* < 0.01). The highest increase was found in *B. cereus* A1-1 (6.06-fold) after being treated for 6 h, whereas an opposite pattern was observed in *V. parahaemolyticus* ATCC17802 (1.77-fold).

As shown in Figure 4C, when compared with the control groups, cell membrane damage rates of all four experimental groups significantly increased (*p* < 0.05), which raised with the increase in treatment time. Significant damage was observed in *B. cereus* A1-1 (2.95-fold) and *V. parahaemolyticus* B4-10 (2.21-fold) after being treated for 2 h, whereas a similar change was found in the other two strains treated for 4 h. Moreover, cell membrane damage of *B. cereus* A1-1 was the most severe among the four strains after being treated for 6 h (8.54-fold).

Taken together, these results demonstrated that the CC 1 from *R. madaio* Makino significantly increased bacterial cell surface hydrophobicity and membrane permeability and decreased membrane fluidity of *V. parahaemolyticus* ATCC17802, *V. parahaemolyticus* B4-10, *V. alginolyticus* ATCC17749, and *B. cereus* A1-1, consistent with the observed bacterial surface structure by the TEM analysis. The damaged cell surface and membrane structure integrity were beneficial for the CC1 to penetrate bacterial cell envelope to target intracellular processes.

### 2.5. Identification of Potential Antibacterial Compounds in the CC 1 from R. madaio Makino

The obtained CC 1 resolved in H_2_O was subjected to UHPLC-MS analysis. As shown in Table 3, a total of 58 different compounds were identified. The highest percentage of these compounds in the CC 1 was p-phenol ethanolamine (18.62%), followed by D-2-aminobutyric acid (9.46%), sucrose (7.01%), turanose (7.01%), and lactulose (7.01%). Some compounds with lower concentrations were also identified from the extract (0.83–0.07%), including a galactose 1-phosphate, L-glutamic acid, and kojibiose (Table 3). Phenols and organic acids have good antioxidant and antibacterial activities [11], while alkaloids can inhibit the formation of and/or disperse bacterial biofilms [12]. For example, the indole of alkaloids is a versatile heterocyclic compound with various pharmacological activities such as anticancer, anticonvulsant, antimicrobial, antitubercular, antimalarial, antiviral, antidiabetic and other miscellaneous activities. Indole also regulates various aspects of bacterial physiology, including spore formation, plasmid stability, resistance to drugs, biofilm formation and virulence [13]. Saccharides have been used to preserve foods for a long history by changing cell osmolarity to inhibit harmful bacterial growth. Kojibiose is a natural disaccharide comprising two glucose moieties linked by an α-1,2 glycosidic bond. It has been reported that Kojibiose can inhibit bacterial proliferation and have anti-inflammatory and antiviral activities [14,15]. In contrast, the certain content of the identified amino acids may not contribute to the observed antibacterial activity by the CC 1 from *R. madaio* Makino.

### 2.6. Differential Transcriptomes Mediated by the CC 1 from R. madaio Makino

To gain insights into the genome-wide gene expression changes mediated by the CC 1 from *R. madaio* Makino, we determined transcriptomes of the four bacterial strains treated for 6 h using Illumina RNA sequencing technology. A complete list of DEGs in the four strains was available in the NCBI SRA database (https://submit.ncbi.nlm.nih.gov/subs/bioproject/, accessed on 17 October 2021) under the accession number PRJNA767551. To validate the transcriptome data, we examined 32 representative DEGs (Appendix A) by RT-qPCR analysis, and the resulting data were correlated with those yielded from the transcriptome analysis (Appendix A).

#### 2.6.1. The Major Altered Metabolic Pathways in *V. alginolyticus* ATCC17749

Approximately 6.73% (316/4698) of *V. alginolyticus* ATCC17749 genes were expressed differently in the experimental group compared with the control group. Among these, 238 genes showed higher transcription levels (FC ≥ 2.0), and 78 genes were down-regulated (FC ≤ 0.5). Based on the comparative transcriptomic analyses, 11 significantly changed metabolic pathways were identified, including valine, leucine and isoleucine degradation; nitrogen, histidine, tryptophan, glyoxylate and dicarboxylate metabolisms; quorum sensing (QS); lysine degradation; fatty acid degradation; amino sugar and nucleotide sugar metabolism; ABC transporters; and mitogen-activated protein kinase (MAPK) signal pathway (Figure 7).

Remarkably, approximately 60 DEGs involved in 10 changed metabolic pathways were significantly up-regulated in *V. alginolyticus* ATCC17749 (2.002- to 87.807-fold) (*p* < 0.05) (Table 4). For example, in the valine, leucine and isoleucine degradation, expression of nine DEGs were significantly up-regulated at the transcription level (2.117- to 4.619-fold) (*p* < 0.05); six DEGs encoding key enzymes in the histidine metabolism were also significantly up-regulated (2.001- to 3.187-fold) (*p* < 0.05); similarly, in the tryptophan metabolism, expression of three DEGs were significantly enhanced (2.123- to 5.154-fold) (*p* < 0.05); additionally, in the lysine degradation, expression of a transcriptional regulator (*N646_3623*) and an arginine/lysine/ornithine decarboxylase (*N646_1979*) were significantly up-regulated (2.972- to 3.332-fold) (*p* < 0.05). These four pathways are related to amino acid degradation metabolisms.

Meanwhile, eight DEGs in the nitrogen metabolism were also significantly up-regulated (2.193- to 87.807-fold) (*p* < 0.05), in which, specifically, one DEG encoding a hydroxylamine reductase (*N646_0236*) was greatly enhanced to express (87.807-fold).

ABC transporters are ATP-dependent efflux transporters to transport lipids, metabolites, exogenous substances and other small molecules out of the cell [16]. They are also the main type of transporters associated with bacterial multidrug resistance [17]. In this study, comparative transcriptome analysis revealed 23 DEGs in ABC transporters and QS that were significantly up-regulated in *V. alginolyticus* ATCC17749 (2.104- to 7.585-fold) (*p* < 0.05) (Table 4). ABC transporter can also catalyze the turnover of lipids in the lipid bilayer that play a critical role in the occurrence and functional maintenance of the cell membrane [18]. In this study, the up-regulated expression of these DEGs suggested that the treatment with the CC 1 from *R. madaio* Makino enhanced the bacterial pumping of exogenous and endogenous metabolites to eliminate cell damage.

In contrast, all DEGs in the MAPK signaling pathway were significantly inhibited (0.123- to 0.369-fold) (*p* < 0.05) (Table 4), which likely led to a highly toxic reactive oxygen species (ROS) accumulation and cell damage.

#### 2.6.2. The Major Altered Metabolic Pathways in *V. parahaemolyticus* ATCC17802

Approximately 19.62% (917/4,674) of *V. parahaemolyticus* ATCC17802 genes were expressed differently in the experimental group compared with the control group. Among these, 128 genes showed higher transcription levels (FC ≥ 2.0), and 789 genes were down-regulated (FC ≤ 0.5). Comparative transcriptome analyses revealed 20 significantly changed metabolic pathways, including methane, nitrogen, glycerolipid, propanoate, sulfur, starch and sucrose, taurine and hypotaurine, phosphonate and phosphinate, and biotin metabolisms; glucagon, and hypoxia inducible factor-1 (HIF-1) signaling pathway; benzoate and ethylbenzene degradation; glycolysis/gluconeogenesis; flagellar assembly; apoptosis; bacterial chemotaxis; cationic antimicrobial peptide (CAMP) resistance; necroptosis, and RNA transport (Figure 8).

Notably, approximately 77 DEGs involved in 12 changed metabolic pathways were significantly down-regulated (0.05- to 0.491-fold) (*p* < 0.05) (Table 5). For example, in the glycolysis/gluconeogenesis, except for an up-regulated 2-oxo acid dehydrogenase subunit E2 (*VP_RS18295*), the other seven DEGs were significantly down-regulation (0.087- to 0.433-fold) (*p* < 0.05); in the propanoate metabolic pathway, express of four DEGs were significantly depressed (0.051- to 0.240-fold) (*p* < 0.05); in the starch and sucrose metabolisms, except for a 4-alpha-glucono transfer (*VP_RS22910*), the other five DEGs were significantly down-regulated (0.206- to 0.499-fold) (*p* < 0.05). These three metabolic pathways were related to carbohydrate metabolisms. Their overall down-regulation trend indicated inactive carbon source transportation and/or utilization, which likely resulted in insufficient energy supply.

Approximately 44 DEGs involved in six energy metabolism pathways in *V. parahaemolyticus* ATCC17802 were also significantly inhibited (*p* < 0.05). For example, the DEG encoding a pyruvate dehydrogenase complex dihydrolipoyllysine-residue acetyltransferase (*VP_RS12210*) was significantly down-regulated (0.331-fold), which connects glycolysis with tricarboxylic acid cycle (TCA) and plays a key role in glucose metabolism [19]. The down-regulation of this enzyme led to a decrease in ATP production and insufficient energy supply [20], which consequently affected bacterial growth and mobility.

The bacterial flagellum is a complex mobility machine with a diversity of roles in pathogenesis, including attachment, colonization, invasion, maintenance and post-infection dispersal in the host [21,22]. In this study, expression of 23 DEGs involved in three substructures of the flagellum, including the filament, hook and basal body [23], were significantly down-regulated at the transcriptional level in *V. parahaemolyticus* ATCC17802 (0.055- to 0.49-fold) (*p* < 0.05), which indicated the depressed flagellum assembly that led to inactive motility of *V. parahaemolyticus* ATCC17802. The 17 down-regulated DEGs in the bacterial chemotaxis [24] (0.101- to 0.491-fold) (*p* < 0.05) provided indirect evidence for this result.

Interestingly, 23 DEGs encoding type III secretory system (T3SS) components were also significantly down-regulated (0.055- to 0.490 -fold) (*p* < 0.05). T3SS enables pathogenic bacteria to directly inject effector proteins into host cells, facilitating bacterial colonization in the host [25]. This result suggested that the cytotoxicity of *V. parahaemolyticus* ATCC17802 was significantly reduced after being treated with the CC 1 from *R. madaio* Makino.

Additionally, in the cationic antimicrobial peptide (CAMP) resistance system, five DEGs were significantly inhibited (0.120- to 0.489-fold), including a multidrug efflux RND transporter permease subunit VmeD (*VP_RS00200*), a thiol: disulfide interchange protein DsbA/DsbL (*VP_RS21260*), an ATP-binding cassette domain-containing protein (*VP_RS05670*), a multidrug efflux RND transporter periplasmic adaptor subunit VmeC (*VP_RS00205*), and a phosphoethanolamine-lipid A transferase (*VP_RS21300*) (Table 5). These results indicated poor efficiency of multidrug efflux transport in *V. parahaemolyticus* ATCC17802 after being treated by the CC 1.

In contrast, five DEGs were significantly up-regulated (2.030- to 4.705-fold), e.g., a response regulator (*VP_RS14060*) and an envelope stress sensor histidine kinase CpxA (*VP_RS14065*) (Table 5).

#### 2.6.3. The Major Altered Metabolic Pathways in *V. parahaemolyticus* B4-10

Approximately 16.75% (783/4674) of *V. parahaemolyticus* B4-10 genes were expressed differently in the experimental group when compared with the control group. Among these genes, 204 showed higher transcription levels (FC ≥ 2.0), and 579 genes were down-regulated (FC ≤ 0.5). Based on the comparative transcriptome analysis, five significantly changed metabolic pathways were identified, including styrene degradation, nitrogen metabolism, QS, folate biosynthesis, and histidine metabolism (Figure 9).

Similar to *V. alginolyticus* ATCC17749, the expression of 10 DEGs in the nitrogen metabolism were significantly up-regulated (2.129- to 107.754-fold) (*p* < 0.05) (Table 6). Notably, one DEG encoding a hydroxylamine reductase (*VP_RS05780*) was greatly up-regulated (107.754-fold). This enzyme can reduce hydroxylamine analogs such as methylhydroxylamine and hydroxyquinone as a scavenger of potentially toxic by-products of nitrate metabolism [26]. Moreover, in the histidine metabolism, four DEGs were highly up-regulated (5.106- to 10.231-fold) (Table 6). The enhanced nitrogen metabolism may have supplemented the energy supply in *V. parahaemolyticus* B4-10 after being treated by the CC 1.

#### 2.6.4. The Major Altered Metabolic Pathways in *B. cereus* A1-1

Approximately 12.57% (720/5730) of *B. cereus* A1-1 genes were expressed differently in the experimental group. Among these genes, 178 showed higher transcription levels (FC ≥ 2.0), and 542 genes were down-regulated (FC ≤ 0.5). The comparative transcriptome analysis revealed 17 significantly changed metabolic pathways, including flagellar assembly; bacterial chemotaxis; two-component system (TCS); thiamine and nitrogen metabolisms; ABC transporters; arginine biosynthesis; fatty acid degradation; alanine, aspartate and glutamate metabolism; riboflavin metabolism; HIF-1 signaling pathway; glycolysis/gluconeogenesis; butanoate, pyrimidine, and propanoate metabolisms; benzoate degradation; and inositol phosphate metabolism (Figure 10).

Similar to the other bacterial strains tested, expression of 12 DEGs involved in the nitrogen metabolism and riboflavin metabolism were significantly up-regulated in *B. cereus* A1-1 (3.325- to 150.780-fold) (*p* < 0.05) (Table 7). Specifically, the DEG encoding a hydroxylamine reductase (*BCN_RS16540*) was also greatly enhanced to express in *B. cereus* A1-1 (150.780-fold).

Conversely, 69 DEGs involved in the flagellar assembly, bacterial chemotaxis, ABC transporters, and TCS were significantly down-regulated at the transcription level in *B. cereus* A1-1 (0.038- to 0.487-fold) (*p* < 0.05) (Table 7), similar to the other bacterial strains treated with the CC1. For example, in the flagellar assembly, expression of 19 DEGs were significantly depressed (0.038- to 0.438-fold) (*p* < 0.05); 9 DEGs in bacterial chemotaxis were significantly down-regulated (0.063- to 0.474-fold); and expression of 33 DEGs in ABC transporters were significantly inhibited (0.051- to 0.487-fold).

Approximately eight DEGs in the TCSs were significantly down-regulated. TCSs are widespread regulatory systems that can help bacteria to control their cellular functions and respond to a diverse range of stimuli [27]. In this study, in the HIF-1 signaling pathway, the expression of a L-lactate dehydrogenase (*BCN_RS24725*) was also significantly down-regulated (0.191-fold). These results indicated the inhibited signal transduction systems in *B. cereus* A1-1.

Additionally, 17 DEGs in the arginine biosynthesis, thiamine metabolism, and alanine, aspartate and glutamate metabolism were all significantly down-regulated (0.031- to 0.498-fold) (*p* < 0.05) (Table 7), which suggested the inhibited energy metabolism in *B. cereus* A1-1 after being treated by the CC 1 from *R. madaio* Makino.

## 3. Materials and Methods

### 3.1. Bacterial Strains and Culture Conditions

Bacterial strains and culture media used in this study are listed in Appendix A. Bacterial culture media were purchased as described previously [28]. *Vibrio* strains were inoculated in media (pH 8.4–8.5) with 3.0% NaCl, while non-*Vibrios* in media (pH 7.0–7.2) with 1% NaCl [28].

### 3.2. Extraction of Bioactive Substances from R. madaio Makino

*R. madaio* Makino was collected in Lishui City (27°25′37″ N, 118°41′28″ E), Zhejiang Province, China in September of 2020. A 500 g of fresh leaf and stem tissues of *R. madaio* Makino was washed clean, dried at room temperature, and then freeze-dried using ALPHA 2-4 LD Plus Freeze Dryer (Martin Christ, Osterode, Germany) at −80 °C for 48 h. The freeze-dried material was crushed using FW-135 High-Speed Crusher (Beijing Kangtuo Medical Instruments Co., Ltd., Beijing, China) and passed through 300 mesh screen. Then, 10.0 g of the powder was mixed with 99-mL chloroform: methanol (2:1, *v/v*, analytical grade, Merck KGaA, Darmstadt, Germany) at a solid to liquid ratio of 1.10 (*m/v*) for 5 h [29]. A 60 mL of H_2_O (Analytical grade, Merck KGaA, Darmstadt,,Germany) was then added, fully mixed, and then sonicated using Scientz IID ULtrasonic Cell Crusher (SCIENT Z, Ningbo, China) at the following parameters: power: 300 W; ultrasonic on time: 1 s; ultrasonic off time: 1 s; working time: 20 min; and probe size: 6 mm. The sonicated mixture was filtered through 20–25 μm membrane (Shanghai Sangon Biological Engineeing Technology and Service Co., Ltd., Shanghai, China), and the filtration was collected for the secondary extraction. The methanol phase was separated from the chloroform phase and then individually evaporated, concentrated on pasting using Rotary Evaporator (IKA, Staufen, Germany).

### 3.3. Antimicrobial Susceptibility Assay

Susceptibility of bacterial strains (Appendix A) to the extracts from *R. madaio* Makino was determined according to the method issued by Clinical and Laboratory Standards Institute (CLSI) (2018, CLSI, M100-S23) using Mueller-Hinton (M-H) agar (CM337) and Mueller-Hinton broth (M391) (OXOID, Basingstoke, UK). Briefly, a 10 μL of crude extracts (500 μg/mL) was added onto each blank disc (6 mm, OXOID, Basingstoke, UK) on MH ager plates. The gentamicin disc (10 μg, OXOID, Basingstoke, UK) was used as a positive control, while the methanol-phase with water and chloroform-phase with ethanol was a negative control, respectively. The plates were incubated at 37 °C for 12 h. Bacteriostatic activity was evaluated by measuring diameters of bacteriostatic circles.

Broth dilution testing (microdilution) (2018, CLSI, M100-S18) was used to determine MICs of the extracts. Briefly, a 100 μL/well of the extracts (1024 μg/mL) was serially diluted, mixed with 100 μL/well of Mueller-Hinton broth (CM337) and 10 μL/well of bacteria strain (1.5 × 10^6^ colony-forming unit (CFU)/mL), and then incubated at 37 °C for 12 h [30]. The MIC was defined as the lowest concentration of a particular antibacterial agent that inhibits bacterial growth (2018, CLSI, M100-S18). The standard solution of gentamicin (100 μg/mL) was purchased from National Standard Material Information Center, Beijing, China.

### 3.4. Prep-HPLC Analysis

Aliquots (10 mg/mL) of freeze-dried samples resolved in H_2_O (Analytical grade, Merck KGaA, Darmstadt, Germany) were centrifuged at 12,000 rpm for 20 min. The supernatant was filtered through 0.22 µm membrane (Sangon, Shanghai, China), and the filtration was collected for further analysis. Prep-HPLC was run using Waters 2707 (Waters, Milford, Massachusetts, USA) linked with UPLC Sunfire C18 column (5 μm, 10 × 250 mm) (Waters, Massachusetts, USA) at the following parameters: column temperature, 40 °C; injection volume, 100 μL; and mobile phase of methanol (eluent A) and water (eluent B) at a flow rate of 4 mL/min (isocratic elution: 0–15 min, 20% eluent A and 80% eluent B). Photo-diode array (PDA) spectra were measured in the wavelength ranging from 200 to 600 nm.

### 3.5. UHPLC–MS Analysis

The UHPLC–MS analysis was carried out using EXIONLC System (Sciex, Framingham, MA, USA) by Shanghai Hoogen Biotech, Shanghai, China using the parameters as described previously [31]. The mobile phase A contained 0.1% formic acid in H_2_O (*v/v*), and mobile phase B was acetonitrile (Merck KGaA, Darmstadt, Germany); column temperature: 40 °C; auto-sampler temperature: 4 °C; injection volume: 2 μL. Typical ion source parameters were: IonSpray voltage: +5500/−4500 V; curtain gas: 35 psi; temperature: 400 °C; ion source Gas 1:60 psi; ion source Gas 2: 60 psi; and declustering potential (DP): ±100 V. The SCIEX Analyst Work Station Software (Version 1.6.3) was employed for multiple reaction monitoring (MRM) data acquisition and processing. In-house R program and database were applied for peak detection and annotation (Shanghai Hoogen Biotech, Shanghai, China).

### 3.6. Transmission Electron Microscope (TEM) Assay

Samples for TEM analysis were prepared according to the method described previously [32]. Briefly, 1 × MIC concentration of CC 1 from *R. madaio* Makino was added in bacterial culture (5 mL) at middle logarithmic growth phase (mid-LQP), and incubated at 37 °C for 2 h, 4 h and 6 h, respectively. A 1.5 mL of the cell suspension were collected, washed, fixed, and observed using SU5000 transmission electron microscope (Hitachi, Tokyo, Japan, 5.0 kV, ×30,000) [32].

### 3.7. Bacterial Cell Surface Hydrophobicity, Membrane Fluidity and Damage Assays

Bacterial cell surface hydrophobicity and membrane fluidity were measured according to the methods by Krausova et al. [33] and Kuhry et al. [34], respectively. In the former method, 1 mL of 98% cetane (Sangon, Shanghai, China) was added into 1 mL of bacterial cell suspension (OD_600 nm_ values of 0.55 to 0.60) and rotated for 1 min and then stood at room temperature for 30 min. The absorbance of the aqueous phase was measured at OD_600 nm_ using BioTek Synergy 2 (BioTek, Burlington, VT, USA). To measure the membrane fluidity, a 200 μL/well of bacterial suspension was mixed with 2 μL of 10 mM 1,6-diphenyl-1,3,5-hexatriene (DPH) (Sangon, China), and the change of fluorescence intensity of each well was measured at excitation light wavelength of 362 nm and emission light wavelength of 427 nm using BioTek Synergy 2 (BioTek, Burlington, VT, USA).

Cell membrane damage was examined according to the method described previously [32]. Briefly, the bacterial cell suspension was double-dyed using propidium iodide (PI, 10 mM final concentration) (Sangon, China), and 5(6)-carboxydiacetate fluorescein succinimidyl ester (CFDA, 10 mM final concentration) (Beijing Solarbio Science & Technology Co. Ltd., Beijing, China), and determined using Flow Cytometer BD FACSVerse™ (Becton, Dickinson and Company, Franklin Lakes, NJ, USA) [32].

### 3.8. Cell Membrane Permeability Analysis

Bacterial culture at the mid-LGS was mixed with 1 × MIC concentration of the CC 1 from *R. madaio* Makino and then incubated at 37 °C for 2 h, 4 h and 6 h. Outer membrane permeability was measured according to the method described previously [35]. Briefly, a 200 μL/well of bacterial cell suspension was mixed with 2 μL/well of 10 mm NPN solution (Sangon, Shanghai, China). The excitation and emission wavelengths were set at 350 nm and 420 nm, respectively, and recorded using BioTek Synergy 2 (BioTek, Burlington, VT, USA) [35].

Inner membrane permeability was measured according to the method described previously [36]. Briefly, a 200 μL/well of bacterial cell suspension was mixed with 2.5 μL/well of 10 mm ONPG solution (Sangon, Shanghai, China). The cell mixture was incubated at 37 °C and measured for each well at OD_415 nm_ using BioTek Synergy 2 (BioTek, Burlington, VT, USA) every 30 min for 5 h, which was marked as OD_1_, while OD_2_ generated from the untreated bacterial suspension was used as a negative control [36].

### 3.9. Illumina RNA Sequencing

Bacterial culture at the mid-LGP was treated with 1 × MIC concentration of the CC 1 from *R. madaio* Makino for 6 h. Total RNA was prepared using RNeasy Protect Bacteria Mini Kit (QIAGEN Biotech Co. Ltd., Frankfurt, Germany) and QIAGEN RNeasy Mini Kit (QIAGEN). DNA was removed from the samples using RNase-Free DNase Set (QIAGEN). Three independently prepared RNA samples were used for each Illumina RNA-sequencing analysis. Illumina sequencing was conducted by Shanghai Majorbio Bio-pharm Technology Co. Ltd. (Shanghai, China) using Illumina HiSeq 2500 platform (Illumina, Santiago, CA, USA). High quality reads that passed the Illumina quality filters were used for sequence analyses [32].

### 3.10. Reverse Transcription Real Time-Quantitative PCR (RT-qPCR) Assay

Total RNA extraction, reverse transcription reactions, and relative quantitative PCR reactions were performed using the same kits and instrument according to the method described previously [31]. The 16S rRNA gene was used as the internal reference gene, and 2^−ΔΔCt^ method was used to calculate relative expression of genes. Oligonucleotide primers used for the RT-qPCR were synthesized by Sangon, Shanghai, China.

### 3.11. Data Analysis

Expression of each gene was calculated using RNA-Seq by Expectation-Maximization (RSEM, http://deweylab.github.io/RSEM/, accessed on 17 October 2021). Genes with the criteria, fold-changes ≥ 2.0 or ≤0.5, and *p*-values < 0.05 relative to the control were defined as DEGs. These DEGs were used for gene set enrichment analysis (GSEA) against the Kyoto Encyclopedia of Genes and Genomes (KEGG) database (https://www.genome.jp/kegg/, accessed on 17 October 2021). Significantly changed GSEA were identified when the enrichment test *p*-value fell below 0.05 [32]. All tests were performed in triplicates. The data were analyzed using SPSS statistical analysis software version 17.0 (SPSS Inc., Armonk, NY, USA).

## 4. Conclusions

In this study, we identified, for the first time, antibacterial components and action modes of methanol-phase extract from one edible herbaceous plant *R. madaio* Makino. The bacteriostatic rate of the extract was 75% against 23 species of common pathogenic bacteria, which was higher than that of the chloroform-phase extract (39%). The methanol-phase extract was further purified using the Prep-HPLC technique, and five separated CCs were obtained. Among these, the CC 1 from *R. madaio* Makino significantly increased bacterial cell surface hydrophobicity and membrane permeability and decreased membrane fluidity of Gram-positive and Gram-negative pathogens, such as *V. parahaemolyticus* ATCC17802, *V. parahaemolyticus* B4-10, *V. alginolyticus* ATCC17749, and *B. cereus* A1-1. The damaged cell surface and membrane structure integrity facilitated the CC1 to penetrate bacterial cell envelope to target intracellular processes. A total of 58 different compounds in the extract were identified using UHPLC–MS technique. Comparative transcriptomic analyses revealed a number of differentially expressed genes (DGEs) and various changed metabolic pathways mediated by the CC1 action, such as down-regulation of carbohydrate transport and/or utilization, and energy metabolism; upward regulation of amino acid and fatty acid degradation, and nitrogen metabolism; and inactive flagellar assembly and mobility in the four bacterial strains. Taken, the results in this study demonstrated that the CC1 from *R. madaio* Makino are promising candidates for antibacterial medicine and human health care products.

## Figures and Tables

**Figure 1 molecules-27-00660-f001:**
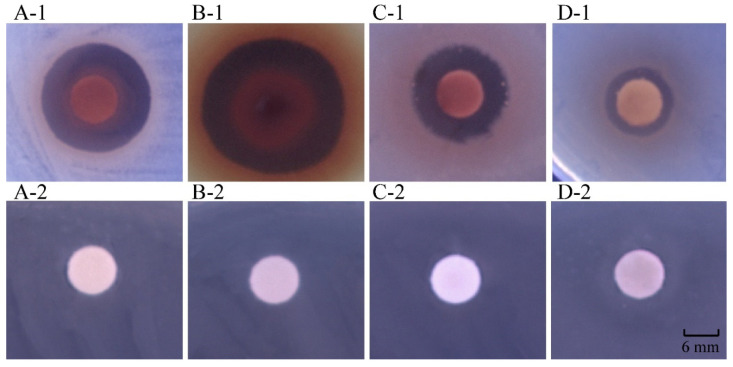
Inhibition activity of the methanol-phase crude extract from *R. madaio* Makino against the four representative bacterial strains. (**A-1**): *B. cereus* A1-1; (**B-1**): *V. alginolyticus* ATCC17749; (**C-1**): *V. Parahaemolyticus* ATCC17802; and (**D-1**): *V. Parahaemolyticus* B4-10. (**A-2**–**D-2**): negative control, respectively.

**Figure 2 molecules-27-00660-f002:**
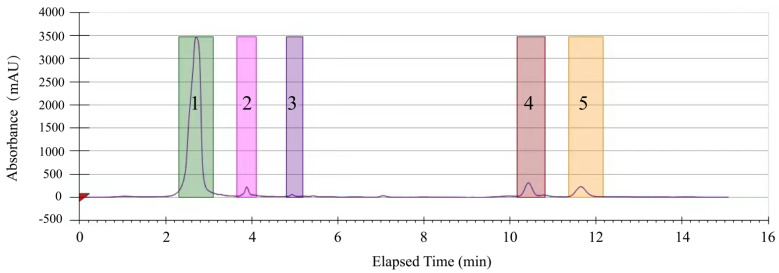
The Prep−HPLC diagram of purifying the methanol-phase crude extract from *R. madaio* Makino.

**Figure 3 molecules-27-00660-f003:**
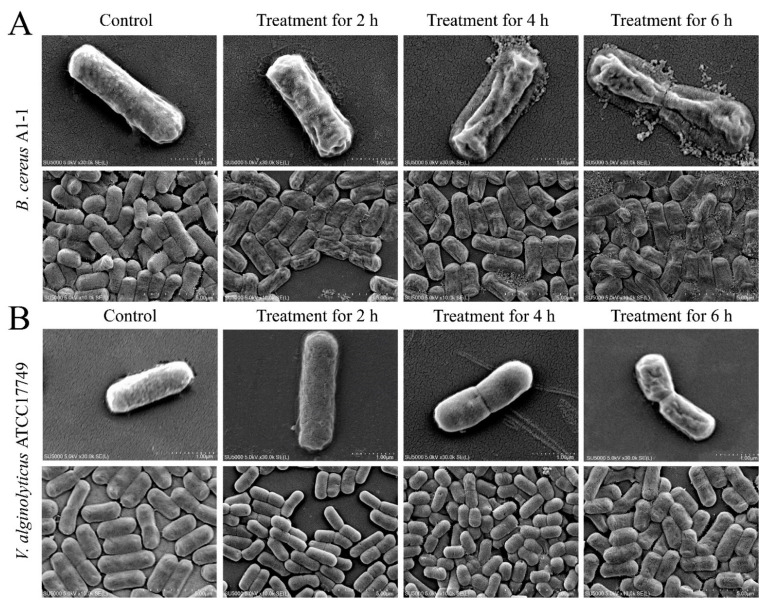
The TEM observation of cell surface structure of the four bacterial strains treated with the CC1 for different times. (**A**): *B. cereus* A1-1; (**B**): *V. alginolyticus* ATCC17749; (**C**): *V. Parahaemolyticus* ATCC17802; and (**D**): *V. Parahaemolyticus* B4-10.

**Figure 4 molecules-27-00660-f004:**
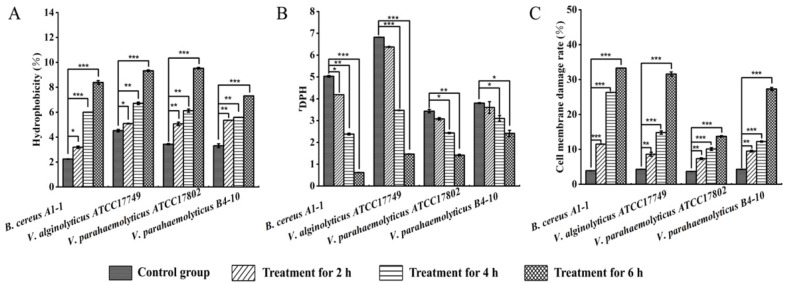
Effects of the CC 1 from *R. madaio* Makino on cell surface hydrophobicity, membrane fluidity and damage of the four bacterial strains. (**A**): cell surface hydrophobicity; (**B**): cell membrane fluidity; and (**C**): cell membrane damage. The results were represented as the mean ± standard deviation of three repetitions. *: *p* < 0.05; **: *p* < 0.01; and ***: *p* < 0.001.

**Figure 5 molecules-27-00660-f005:**
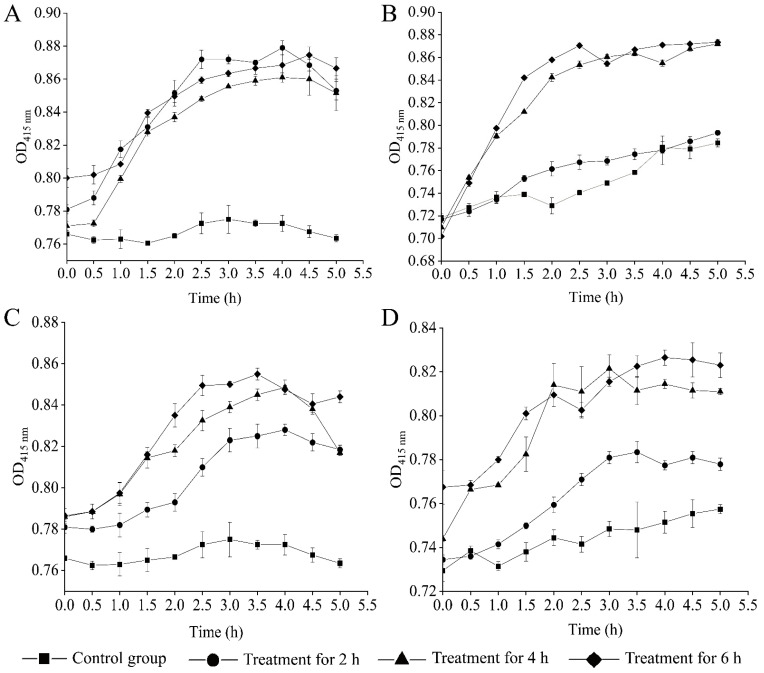
Effects of the CC 1 from *R. madaio* Makino on inner cell membrane permeability of the four bacterial strains. (**A**): *B. cereus* A1-1; (**B**): *V. alginolyticus* ATCC17749; (**C**): *V. Parahaemolyticus* ATCC17802; and (**D**): *V. Parahaemolyticus* B4-10.

**Figure 6 molecules-27-00660-f006:**
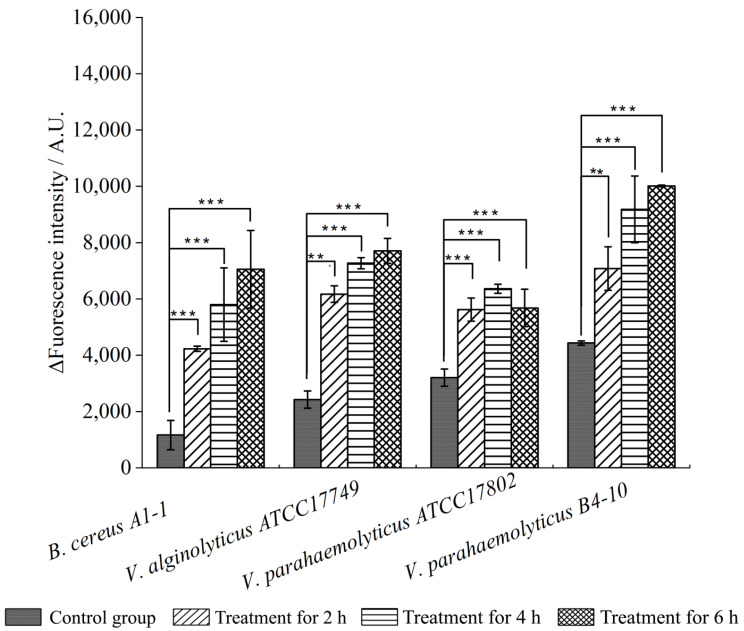
Effects of the CC 1 from *R. madaio* Makino on outer cell membrane permeability of the four bacterial strains. The results were represented as the mean ± standard deviation of three repetitions. **: *p* < 0.01; ***: *p* < 0.001.

**Figure 7 molecules-27-00660-f007:**
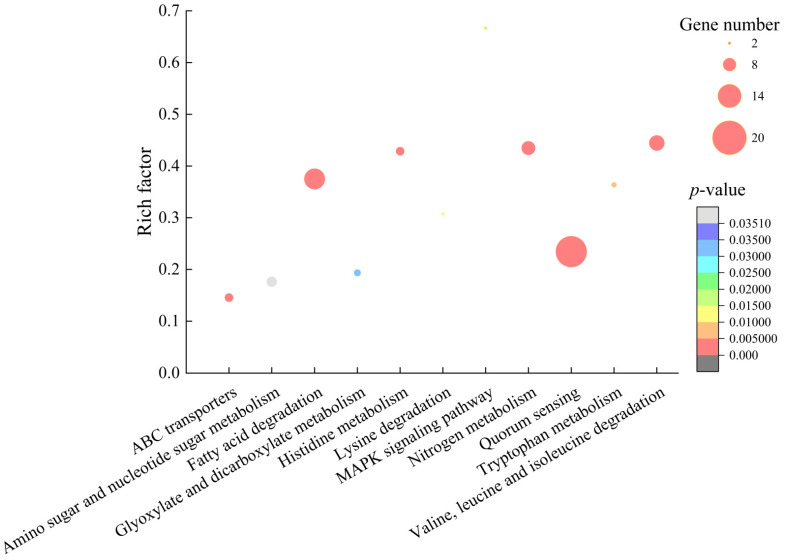
The 11 significantly altered metabolic pathways in *V. alginolyticus* ATCC17749 mediated by the CC 1 from *R. madaio* Makino.

**Figure 8 molecules-27-00660-f008:**
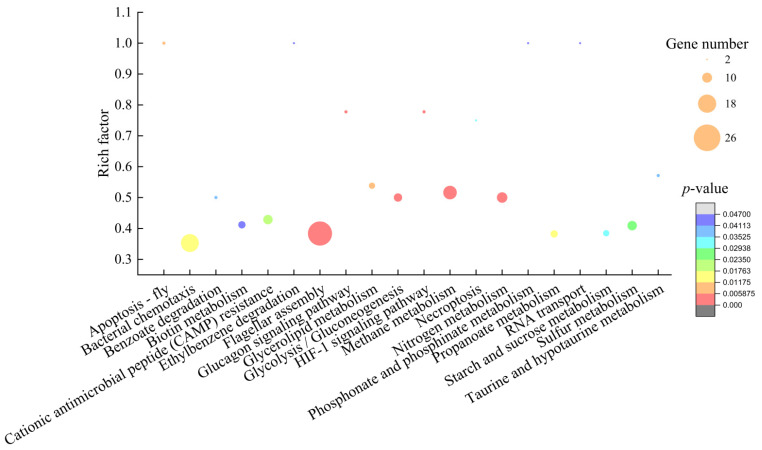
The 20 significantly altered metabolic pathways in *V. parahaemolyticus* ATCC17802 mediated by the CC 1 from *R. madaio* Makino.

**Figure 9 molecules-27-00660-f009:**
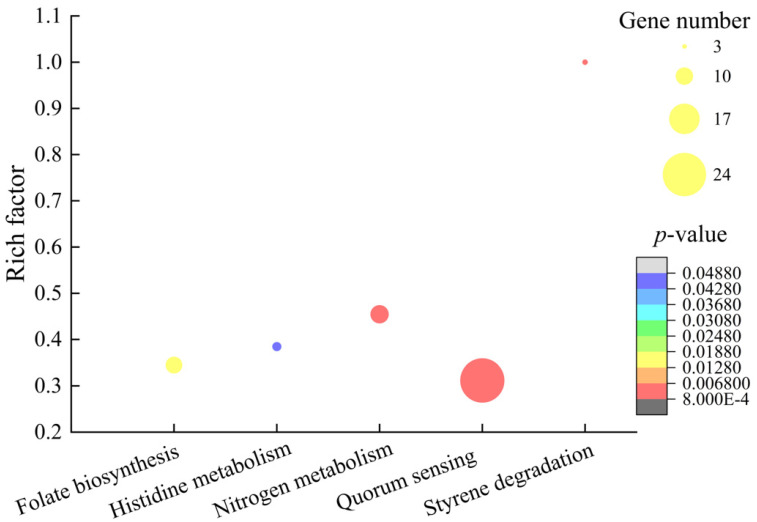
The 5 significantly altered metabolic pathways in *V. parahaemolyticus* B4-10 mediated by the CC 1 from *R. madaio* Makino.

**Figure 10 molecules-27-00660-f010:**
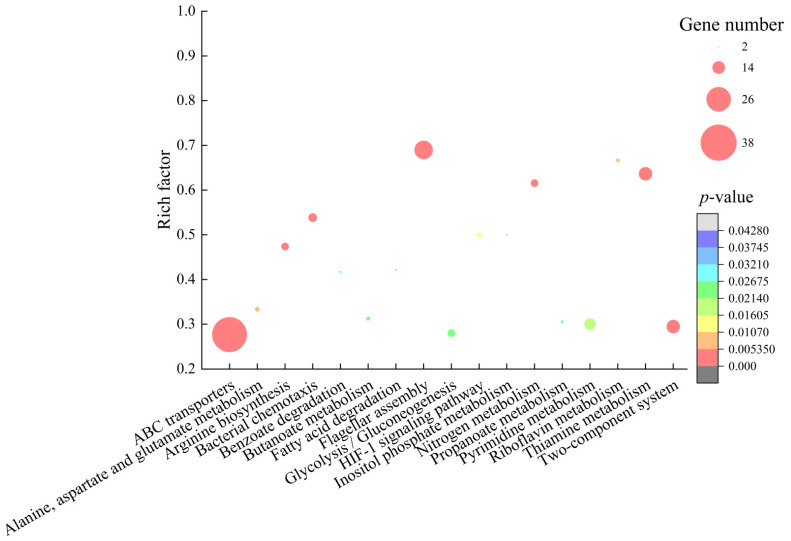
The 17 significantly altered metabolic pathways in *B. cereus* A1-1 mediated by the CC 1 from *R. madaio* Makino.

**Table 1 molecules-27-00660-t001:** Antibacterial activity of crude extracts from *R. madaio* Makino.

pStrain	Inhibition Zone (Diameter, mm)		MIC (μg/mL)	
	CPE	MPE	CPE	MPE
*Aeromonas hydrophila* ATCC35654	—	11.30 ± 0.47	—	126
*Bacillus cereus* A1-1	—	14.70 ± 1.25	—	32
*Enterobacter cloacae* ATCC13047	7.90 ± 0.05	13.00 ± 0.86	512	64
*Enterobacter cloacae*	—	8.30 ± 0.24	—	512
*Escherichia coli* ATCC8739	—	—	—	—
*Escherichia coli* ATCC25922	—	—	—	—
*Escherichia coli* K12	—	9.30 ± 1.25	—	128
*Enterobacter sakazakii* CMCC45401	8.90 ± 0.14	8.70 ± 0.47	256	512
*Listeria monocytogenes* ATCC19115	9.80 ± 0.17	—	256	—
*Pseudomonas aeruginosa* ATCC9027	—	9.30 ± 0.94	—	256
*Pseudomonas aeruginosa* ATCC27853	—	9.00 ± 0.21	—	256
*Salmonella choleraesuis* ATCC13312	—	9.70 ± 0.94	—	256
*Salmonella paratyphi-A* CMCC50093	8.70 ± 0.94	9.40 ± 0.43	512	256
*Salmonella typhimurium* ATCC15611	8.90 ± 0.17	14.00 ± 0.82	256	32
*Salmonella*	8.20 ± 0.17	20.30 ± 0.47	512	8
*Shigella dysenteriae* CMCC51252	—	—	—	—
*Shigella flexneri* CMCC51572	—	10.00 ± 0.00	—	128
*Shigella flexneri* ATCC12022	—	—	—	—
*Shigella flexneri* CMCC51574	—	—	—	—
*Shigella sonnei* ATCC25931	—	—	—	—
*Shigella sonnet* CMCC51592	9.40 ± 0.29	8.10 ± 0.05	256	512
*Staphylococcus aureus* ATCC25923	10.60 ± 0.42	8.10 ± 0.29	128	512
*Staphylococcus aureus* ATCC8095	8.00 ± 0.05	7.30 ± 0.21	512	1024
*Staphylococcus aureus* ATCC29213	—	7.20 ± 0.08	—	1024
*Staphylococcus aureus* ATCC6538	10.00 ± 0.82	10.00 ± 2.16	256	256
*Staphylococcus aureus* ATCC6538P	—	10.50 ± 0.41	—	128
*Staphylococcus aureus*	7.00 ± 0.00	8.50 ± 0.41	1024	512
*Vibrio alginolyticus* ATCC17749	—	24.30 ± 1.25	—	4
*Vibrio alginolyticus* ATCC33787	—	—	—	—
*Vibrio cholerae* Q10-54	—	—	—	—
*Vibrio cholerae* b10-49	—	9.00 ± 0.24	—	256
*Vibrio cholerae* GIM1.449	10.30 ± 0.36	10.50 ± 0.41	256	128
*Vibrio fluvialis* ATCC33809	11.30 ± 0.47	7.90 ± 0.09	128	512
*Vibrio harvey* ATCC BAA-1117	—	8.00 ± 0.05	—	512
*Vibrio harveyi* ATCC33842	—	—	—	—
*Vibrio metschnikovii* ATCC700040	8.40 ± 0.42	—	512	—
*Vibrio mimicus* bio-56759	9.20 ± 0.12	13.00 ± 0.82	512	64
*Vibrio parahaemolyticus* B3-13	10.50 ± 0.41	9.10 ± 0.12	128	256
*Vibrio parahaemolyticus* B4-10	—	10.30 ± 0.47	—	128
*Vibrio parahaemolyticus* B5-29	—	12.30 ± 0.94	—	64
*Vibrio parahaemolyticus* B9-35	—	8.30 ± 0.21	—	512
*Vibrio parahaemolyticus* ATCC17802	—	13.70 ± 0.94	—	128
*Vibrio parahaemolyticus* ATCC33847	—	13.00 ± 0.00	—	64
*Vibrio vulnificus* ATCC27562	11.70 ± 1.25	8.70 ± 0.47	128	256

Note: CPE: chloroform phase extract. MPE: methanol phase extract. —: no bacteriostasis activity. Inhibition zone: diameter includes the disk diameter (6 mm). MIC: minimum inhibitory concentration. Values are means ± S.D. of three parallel measurements.

**Table 2 molecules-27-00660-t002:** Antibacterial activity of the CC 1 from *R. madaio* Makino.

Strain	Inhibition Zone (Diameter, mm)	MIC (μg/mL)
*B. cereus* A1-1	10.30 ± 0.24	128
*S. typhimurium* ATCC15611	7.90 ± 0.22	512
*S. aureus* ATCC6538	7.00 ± 0.05	1024
*V. alginolyticus* ATCC17749	11.20 ± 0.21	64
*V. parahaemolyticus* ATCC17802	11.10 ± 0.08	64
*V. parahaemolyticus* ATCC33847	7.90 ± 0.25	256
*V. parahaemolyticus* B3-13	7.10 ± 0.09	512
*V. parahaemolyticus* B4-10	9.40 ± 0.26	256
*V. parahaemolyticus* B5-29	8.10 ± 0.12	512

Note: MIC: minimum inhibitory concentration.

**Table 3 molecules-27-00660-t003:** Compounds identified in the CC 1 from *R. madaio* Makino by the UHPLC–MS analysis.

Peak No.	Identified Compound	Compound Nature	Rt (min)	Formula	Exact Mass	Peak Area (%)
1	*p*-Octopamine	Biogenic amine	3.84	C_8_H_11_NO_2_	153.08	18.62
2	D-alpha-Aminobutyric acid	Amino acids and derivatives	0.65	C_4_H_9_NO_2_	103.06	9.46
3	Sucrose	Carbohydrates	0.89	C_12_H_22_O_11_	342.12	7.01
4	Turanose	Carbohydrates	0.79	C_12_H_22_O_11_	342.12	7.01
5	Lactulose	Organooxygen compounds	0.77	C_12_H_22_O_11_	342.12	7.01
6	L-Arginine	Amino acids and derivatives	0.60	C_6_H_14_N_4_O_2_	174.11	4.98
7	L-Lysine; L-Glutamine	Amino acids and derivatives	0.64	C_6_H_14_N_2_O_2_	146.11	4.68
8	D-Glutamine	Amino acids and derivatives	0.66	C_5_H_10_N_2_O_3_	146.07	4.68
9	(2*E*)-Decenoyl-ACP	Carboxylic acids and derivatives	1.47	C_6_H_11_NO_2_	129.08	3.14
10	O-Acetylethanolamine	Alkaloids	0.67	C_4_H_9_NO_2_	103.06	3.00
11	L-Pipecolic acid	Amino acids and derivatives	0.69	C_6_H_11_NO_2_	129.08	2.48
12	Pyrrolidonecarboxylic acid	Amino acids and derivatives	0.67	C_5_H_7_NO_3_	129.04	2.48
13	D-Maltose	Carbohydrates	0.76	C_12_H_22_O_11_	342.12	1.86
14	Trigonelline	Alkaloids	0.82	C_7_H_7_NO_2_	137.05	1.74
15	Indole	Alkaloids	3.82	C_8_H_7_N	117.06	1.66
16	Uridine 5’-diphospho-d-glucose	Carbohydrates	0.71	C_15_H_24_N_2_O_17_P_2_	566.06	1.65
17	Proline; L-Proline	Amino acids and derivatives;	0.73	C_5_H_9_NO_2_	115.06	1.53
18	D-Proline	Amino acids and derivatives	0.76	C_5_H_9_NO_2_	115.06	1.53
19	Lubiprostone	Fatty acyls	12.75	C_20_H_32_F_2_O_5_	390.22	1.40
20	Phosphoric acid	Inganic acids	0.65	H_3_O_4_P	97.98	1.29
21	Sarracine	Alkaloids	13.14	C_18_H_27_NO_5_	337.19	0.83
22	Galactose 1-phosphate	Organooxygen compounds	0.65	C_6_H_13_O_9_P	260.03	0.75
23	L-Glutamic acid	Amino acids and derivatives	0.66	C_5_H_9_NO_4_	147.05	0.67
24	Kojibiose	Carbohydrates	0.72	C_12_H_22_O_11_	342.12	0.50
25	Glucose 6-phosphate	Carbohydrates	0.65	C_6_H_13_O_9_P	260.03	0.49
26	*p*-Aminobenzoate	Benzoic acid derivatives	0.74	C_7_H_7_NO_2_	137.05	0.47
27	Betaine	Alkaloids	1.06	C_5_H_11_NO_2_	117.08	0.47
28	L-Histidine	Amino acids and derivatives	0.59	C_6_H_9_N_3_O_2_	155.07	0.44
29	8,9-DiHETrE	Fatty Acyls	13.03	C_20_H_34_O_4_	338.25	0.43
30	Gluconic acid	Organic acids	0.69	C_6_H_12_O_7_	196.06	0.43
31	*N*,*N*-Dimethylglycine	Amino acids and derivatives	1.04	C_4_H_9_NO_2_	103.05	0.40
32	2-Aminoisobutyric acid	Amino acids and derivatives	0.98	C_4_H_9_NO_2_	103.06	0.37
33	Diallyl disulfide	Organic disulfide	0.68	C_6_H_10_S_2_	146.02	0.37
34	2-Hydroxybutanoic acid	Organic acids	0.64	C_4_H_8_O_3_	104.05	0.35
35	Beta-Sitosterol	Steroids	12.93	C_29_H_50_O	414.39	0.33
36	Phosphorylcholine	Cholines	0.67	C_5_H_14_NO_4_P	183.07	0.31
37	Campesterol	Steroids and steroid derivatives	12.18	C_28_H_48_O	400.37	0.31
38	Gemcitabine	Pyrimidine nucleosides	0.75	C_9_H_11_F_2_N_3_O_4_	263.07	0.30
39	L-Threonine	Amino acids and derivatives	0.64	C_4_H_9_NO_3_	119.06	0.29
40	L-Homoserine	Amino acids and derivatives	0.67	C_4_H_9_NO_3_	119.05	0.29
41	3-Ethyl-1,2-benzenediol	Phenols	0.74	C_8_H_10_O_2_	138.07	0.29
42	Diacylglycerol	Glycerolipids	13.42	C_37_H_70_O_5_	568.51	0.28
43	Rutin	Flavonoids	5.85	C_27_H_30_O_16_	610.15	0.27
44	cis-Aconitic acid	Organic acids and derivatives	1.46	C_6_H_6_O_6_	174.02	0.25
45	L-Citruline	Amino acids and derivatives	0.66	C_6_H_13_N_3_O_3_	175.09	0.25
46	Wighteone	Flavonoids	13.01	C_20_H_18_O_5_	338.11	0.24
47	Beta-d-Fructose 2-phosphate	Carbohydrates	0.75	C_6_H_13_O_9_P	260.03	0.22
48	Maltol	Flavonoids	0.90	C_6_H_6_O_3_	126.03	0.21
49	Itaconic acid	Organic acids	0.52	C_5_H_6_O_4_	130.03	0.21
50	Safrole	Benzodioxoles	12.26	C_10_H_10_O_2_	162.07	0.20
51	22-Dehydroclerosterol	Steroids	12.59	C_29_H_46_O	410.35	0.18
52	8-Hydroxybergapten	Coumarins	10.56	C_12_H_8_O_5_	232.04	0.17
53	Isoquercitrin	Flavonoids	6.06	C_21_H_20_O_12_	464.10	0.14
54	Miltirone	Diterpenoids	12.98	C_19_H_22_O_2_	282.16	0.11
55	Puerarin	Flavonoids	4.89	C_21_H_20_O_9_	416.11	0.11
56	Cinchonine	Alkaloids	11.99	C_19_H_22_N_2_O	294.17	0.09
57	3-Ethoxy-4-hydroxybenzaldehyde	Phenols	5.72	C_9_H_10_O_3_	166.06	0.07
58	Lumichrome	Alkaloids	6.69	C_12_H_10_N_4_O_2_	242.08	0.07

**Table 4 molecules-27-00660-t004:** Major altered metabolic pathways in *V. alginolyticus* ATCC17749 treated by the CC1 from *R. madaio* Makino.

Metabolic Pathway	Gene ID	Fold Change	Gene Description
Valine, leucine and isoleucine degradation	*N646_4585*	2.117	Acetoacetyl-coenzyme A synthetase
	*N646_4506*	2.127	Putative 3-hydroxyisobutyrate dehydrogenase
	*N646_4019*	2.293	Acetoacetyl-coenzyme A synthetase
	*N646_4049*	2.793	Putative acyl-CoA carboxyltransferase beta chain
	*N646_4047*	3.123	Putative acyl-CoA carboxylase alpha chain
	*N646_4057*	3.302	3-hydroxyisobutyrate dehydrogenase
	*N646_4048*	4.128	Putative enoyl-CoA hydratase/isomerase
	*N646_4053*	4.602	Putative aldehyde dehydrogenase
	*N646_4050*	4.619	Putative acyl-CoA dehydrogenase
Nitrogen metabolism	*N646_3727*	2.193	Putative oxidoreductase protein
	*N646_4426*	2.656	Hypothetical protein
	*N646_3915*	5.506	Periplasmic nitrate reductase
	*N646_4365*	5.657	Hypothetical protein
	*N646_3914*	6.137	Periplasmic nitrate reductase%2C cytochrome c-type protein
	*N646_4364*	11.868	Nitrite reductase [NAD(P)H]%2C small subunit
	*N646_1010*	29.988	Nitrite reductase periplasmic cytochrome c552
	*N646_0236*	87.807	Hydroxylamine reductase
Quorum sensing	*N646_0372*	2.104	ABC-type spermidine/putrescine transport system%2C permease component II
	*N646_2230*	2.108	Peptide ABC transporter%2C permease protein
	*N646_4026*	2.258	Putative ABC transporter%2C membrane spanning protein
	*N646_1576*	2.315	Peptide ABC transporter%2C periplasmic peptide-binding protein
	*N646_0379*	2.493	Oligopeptide ABC transporter%2C permease protein
	*N646_2228*	2.531	Peptide ABC transporter%2C periplasmic peptide-binding protein
	*N646_4027*	2.666	Putative high-affinity branched-chain amino acid transport permease protein
	*N646_0377*	2.688	Oligopeptide ABC transporter%2C ATP-binding protein
	*N646_1580*	2.821	Peptide ABC transporter%2C ATP-binding protein
	*N646_0378*	2.836	Oligopeptide ABC transporter%2C ATP-binding protein
	*N646_4024*	2.850	Putative high-affinity branched-chain amino acid transport ATP-binding protein
	*N646_0380*	2.854	Oligopeptide ABC transporter%2C permease protein
	*N646_4025*	2.951	Putative long-chain-fatty-acid-CoA ligase
	*N646_0381*	3.075	Oligopeptide ABC transporter%2C periplasmic oligopeptide-binding protein
	*N646_0370*	3.909	Putative ATP-binding component of ABC transporter
	*N646_4029*	4.034	Putative high-affinity branched-chain amino acid transport ATP-binding protein
	*N646_0371*	4.049	Putative permease of ABC transporter
	*N646_0367*	4.112	Putative binding protein component of ABC transporter
Histidine metabolism	*N646_0312*	2.001	Formimidoylglutamase
	*N646_0189*	2.072	Imidazoleglycerol-phosphate dehydratase/histidinol-phosphatase
	*N646_0190*	2.090	Imidazole glycerol phosphate synthase subunit HisH
	*N646_0313*	3.141	Imidazolonepropionase
	*N646_0311*	3.168	Urocanate hydratase
	*N646_0310*	3.187	Histidine ammonia-lyase
Fatty acid degradation	*N646_1753*	0.344	Hypothetical protein
	*N646_0066*	2.033	Amino acid ABC transporter%2C permease protein
	*N646_3145*	2.064	Rubredoxin/rubredoxin reductase
	*N646_2209*	2.122	Acetyl-CoA C-acyltransferase FadA
	*N646_3116*	2.163	Maltose ABC transporter periplasmic protein
	*N646_3117*	2.319	Maltose/maltodextrin ABC transporter%2C ATP-binding protein
	*N646_3389*	2.793	Putative ferrichrome ABC transporter (permease)
	*N646_1395*	2.879	Acyl-CoA dehydrogenase
	*N646_4429*	3.400	Nitrate ABC transporter nitrate-binding protein
	*N646_4028*	5.585	Hypothetical protein
	*N646_4427*	6.398	Hypothetical protein
	*N646_3568*	14.448	Putative ABC transporter%2C ATP-binding protein
ABC transporters	*N646_4485*	2.173	Arginine ABC transporter%2C permease protein
	*N646_4527*	3.899	Putative inner-membrane permease
	*N646_4487*	4.958	Arginine ABC transporter%2C periplasmic arginine-binding protein
	*N646_4488*	5.676	Arginine ABC transporter%2C ATP-binding protein
	*N646_4486*	7.585	ABC-type arginine transport system%2C permease component
Tryptophan metabolism	*N646_2210*	2.123	Fatty oxidation complex%2C alpha subunit
	*N646_3629*	2.155	Tryptophanase
	*N646_4052*	5.154	Putative acyl-CoA thiolase
Lysine degradation	*N646_3623*	2.972	Transcriptional regulator
	*N646_1979*	3.332	Arginine/lysine/ornithine decarboxylase
MAPK signaling pathway	*N646_2909*	0.123	Cation transport ATPase%2C E1-E2 family protein
	*N646_3134*	0.369	Catalase
Glyoxylate and dicarboxylate metabolism	*N646_1965*	2.122	Acetyl-coenzyme A synthetase
	*N646_2741*	2.135	Isocitrate lyase
	*N646_2740*	2.88	Malate synthase
	*N646_3637*	3.006	Malate synthase
Amino sugar and nucleotide sugar metabolism	*N646_4226*	0.400	Glucose-1-phosphate adenylyltransferase
	*N646_1583*	2.322	Beta-*N*-hexosaminidase
	*N646_3834*	2.610	Hypothetical protein
	*N646_1582*	3.440	Ptative *N*-acetylglucosamine kinase
	*N646_4346*	4.386	Ptative mannose-6-phosphate isomerase
	*N646_3455*	5.366	Hpothetical protein

**Table 5 molecules-27-00660-t005:** Major altered metabolic pathways in *V. parahaemolyticus* ATCC17802 treated by the CC1 from *R. madaio* Makino.

Metabolic Pathway	Gene ID	Fold Change	Gene Description
Methane metabolism	*VP_RS15865*	0.091	NapC/NirT family cytochrome c
	*VP_RS15860*	0.067	Trimethylamine-*N*-oxide reductase 2
	*VP_RS07325*	0.224	Acetate kinase
	*VP_RS13930*	0.206	2%2C3-bisphosphoglycerate-independent phosphoglycerate mutase
	*VP_RS18135*	0.104	Formate dehydrogenase subunit gamma
	*VP_RS12615*	0.320	Phosphate acetyltransferase
	*VP_RS07335*	0.227	Trimethylamine-*N*-oxide reductase TorA
	*VP_RS15585*	0.304	S-(hydroxymethyl)glutathione dehydrogenase/class III alcohol dehydrogenase
	*VP_RS05645*	0.302	Phosphoglycerate dehydrogenase
	*VP_RS07330*	0.338	Pentaheme c-type cytochrome TorC
	*VP_RS05030*	0.381	Molecular chaperone TorD
	*VP_RS15580*	0.412	S-formylglutathione hydrolase
	*VP_RS05640*	0.342	6-phosphofructokinase
Glycolysis/Gluconeogenesis	*VP_RS23260*	0.087	6-phospho-beta-glucosidase
	*VP_RS12915*	0.272	6-phospho-beta-glucosidase
	*VP_RS12215*	0.310	Pyruvate dehydrogenase (acetyl-transferring)
	*VP_RS12210*	0.331	Pyruvate dehydrogenase complex dihydrolipoyllysine-residue acetyltransferase
	*VP_RS13410*	0.406	Glucose-6-phosphate isomerase
	*VP_RS10485*	0.416	D-hexose-6-phosphate mutarotase
	*VP_RS09910*	0.433	Pyruvate kinase
	*VP_RS18295*	2.558	2-oxo acid dehydrogenase subunit E2
Flagellar assembly	*VP_RS22540*	0.055	Flagellar biosynthesis protein FliQ
	*VP_RS16540*	0.064	Flagellar basal body rod protein FlgB
	*VP_RS16565*	0.086	Flagellar basal-body rod protein FlgG
	*VP_RS22520*	0.091	OmpA family protein
	*VP_RS16550*	0.129	Flagellar hook assembly protein FlgD
	*VP_RS22605*	0.193	Flagellar motor stator protein MotA
	*VP_RS22545*	0.210	Flagellar biosynthetic protein FliR
	*VP_RS22575*	0.225	Flagellar filament capping protein FliD
	*VP_RS22535*	0.237	Flagellar type III secretion system pore protein FliP
	*VP_RS22490*	0.265	Flagellar protein export ATPase FliI
	*VP_RS16555*	0.272	Flagellar basal body protein FlgE
	*VP_RS22590*	0.281	Flagellar hook-length control protein FliK
	*VP_RS16575*	0.327	Flagellar basal body P-ring protein FlgI
	*VP_RS10920*	0.363	Flagellar M-ring protein FliF
	*VP_RS22495*	0.366	Flagellar assembly protein H
	*VP_RS10900*	0.386	Flagella biosynthesis chaperone FliJ
	*VP_RS16585*	0.396	Flagellar hook-associated protein FlgK
	*VP_RS16590*	0.412	Flagellar hook-associated protein FlgL
	*VP_RS13775*	0.416	Sel1 repeat family protein
	*VP_RS10835*	0.429	RNA polymerase sigma factor FliA
	*VP_RS10895*	0.452	Flagellar hook-length control protein FliK
	*VP_RS03835*	0.462	Flagellar hook protein FlgE
	*VP_RS03855*	0.490	Flagellar basal body P-ring protein FlgI
Glucagon signaling pathway	*VP_RS01720*	0.369	Pyruvate kinase PykF
	*VP_RS18300*	3.294	Alpha-ketoacid dehydrogenase subunit beta
	*VP_RS22915*	5.913	Glycogen/starch/alpha-glucan phosphorylase
HIF-1 signaling pathway	*VP_RS10480*	0.168	Type I glyceraldehyde-3-phosphate dehydrogenase
	*VP_RS14700*	0.301	ArsJ-associated glyceraldehyde-3-phosphate dehydrogenase
	*VP_RS12650*	0.479	Phosphoglycerate kinase
Nitrogen metabolism	*VP_RS20240*	0.126	Nitrite reductase large subunit NirB
	*VP_RS02310*	0.158	Glutamate synthase subunit beta
	*VP_RS20280*	0.226	Nitrate reductase
	*VP_RS02315*	0.236	Glutamate synthase large subunit
	*VP_RS20255*	0.270	ABC transporter substrate-binding protein
	*VP_RS12190*	0.418	Carbonate dehydratase
	*VP_RS20915*	2.061	Nitrate reductase cytochrome c-type subunit
	*VP_RS20910*	2.197	Periplasmic nitrate reductase subunit alpha
	*VP_RS05780*	14.974	Hydroxylamine reductase
	*VP_RS09370*	19.809	Ammonia-forming nitrite reductase cytochrome c552 subunit
Glycerolipid metabolism	*VP_RS01760*	0.040	Dihydroxyacetone kinase ADP-binding subunit DhaL
	*VP_RS01755*	0.067	Dihydroxyacetone kinase subunit DhaK
	*VP_RS21295*	0.193	Diacylglycerol kinase
	*VP_RS11580*	0.239	Glycerol kinase GlpK
	*VP_RS15810*	0.431	Glycerate kinase
	*VP_RS05740*	2.015	Triacylglycerol lipase
Apoptosis	*VP_RS23210*	0.086	Alkyl hydroperoxide reductase subunit C
	*VP_RS20650*	0.282	C-type cytochrome
	*VP_RS02795*	0.415	Peroxiredoxin C
Bacterial chemotaxis	*VP_RS22610*	0.101	OmpA family protein
	*VP_RS22160*	0.243	Methyl-accepting chemotaxis protein
	*VP_RS03815*	0.255	Protein-glutamate O-methyltransferase
	*VP_RS17585*	0.267	Methyl-accepting chemotaxis protein
	*VP_RS22500*	0.294	Flagellar motor switch protein FliG
	*VP_RS22100*	0.337	Methyl-accepting chemotaxis protein
	*VP_RS10915*	0.356	Flagellar motor switch protein FliG
	*VP_RS05760*	0.374	Methyl-accepting chemotaxis protein
	*VP_RS10820*	0.386	Chemotaxis protein CheA
	*VP_RS10825*	0.389	Protein phosphatase CheZ
	*VP_RS10880*	0.411	Flagellar motor switch protein FliN
	*VP_RS03810*	0.415	Chemotaxis protein CheV
	*VP_RS03305*	0.433	Flagellar motor protein PomA
	*VP_RS10815*	0.471	Chemotaxis response regulator protein-glutamate methylesterase
	*VP_RS10830*	0.473	Chemotaxis response regulator CheY
	*VP_RS05310*	0.486	Methyl-accepting chemotaxis protein
	*VP_RS10800*	0.491	Chemotaxis protein CheW
Propanoate metabolism	*VP_RS01750*	0.051	Glycerol dehydrogenase
	*VP_RS04855*	0.072	Formate C-acetyltransferase
	*VP_RS18985*	0.119	Acetyl-CoA carboxylase%2C carboxyltransferase subunit beta
	*VP_RS16405*	0.240	Aspartate aminotransferase family protein
	*VP_RS07930*	2.084	2-methylcitrate synthase
	*VP_RS07925*	2.094	Fe/S-dependent 2-methylisocitrate dehydratase AcnD
	*VP_RS20545*	2.450	CoA-acylating methylmalonate-semialdehyde dehydrogenase
Cationic antimicrobial peptide (CAMP) resistance	*VP_RS00200*	0.120	Multidrug efflux RND transporter permease subunit VmeD
	*VP_RS00205*	0.159	Multidrug efflux RND transporter periplasmic adaptor subunit VmeC
	*VP_RS21260*	0.344	Thiol: disuLfide interchange protein DsbA/DsbL
	*VP_RS05670*	0.456	ATP-binding cassette domain-containing protein
	*VP_RS21300*	0.489	Phosphoethanolamine-lipid A transferase
	*VP_RS05315*	2.030	Multidrug efflux RND transporter periplasmic adaptor subunit VmeA
	*VP_RS20865*	2.560	Multidrug efflux RND transporter periplasmic adaptor subunit VmeY
	*VP_RS14065*	4.124	Envelope stress sensor histidine kinase CpxA
	*VP_RS14060*	4.705	Response regulator
Sulfur metabolism	*VP_RS07020*	0.050	Dimethyl sulfoxide reductase subunit A
	*VP_RS07030*	0.052	Dimethyl sulfoxide reductase anchor subunit
	*VP_RS07025*	0.058	Dimethyl sulfoxide reductase subunit B
	*VP_RS05930*	0.110	Cytochrome subunit of suLfide dehydrogenase
	*VP_RS03905*	0.337	Cysteine synthase A
	*VP_RS13370*	0.417	Assimilatory suLfite reductase (NADPH) hemoprotein subunit
	*VP_RS13375*	0.440	Assimilatory sulfite reductase (NADPH) flavoprotein subunit
	*VP_RS01435*	0.442	Sulfate adenylyltransferase subunit CysN
	*VP_RS01430*	0.450	Sulfate adenylyltransferase subunit CysD
Starch and sucrose metabolism	*VP_RS12920*	0.206	PTS lactose/cellobiose transporter subunit IIA
	*VP_RS19165*	0.393	Glucose-1-phosphate adenylyltransferase
	*VP_RS03410*	0.474	Alpha%2Calpha-phosphotrehalase
	*VP_RS23025*	0.498	Glycogen debranching protein GlgX
	*VP_RS03405*	0.499	PTS trehalose transporter subunit IIBC
	*VP_RS22910*	4.693	4-alpha-glucanotransferase
Necroptosis	*VP_RS04005*	0.261	Molecular chaperone HtpG
	*VP_RS00595*	0.363	Glutamate-ammonia ligase
Taurine and hypotaurine metabolism	*VP_RS10125*	0.167	Acetate kinase
	*VP_RS05370*	0.219	Alanine dehydrogenase
	*VP_RS10130*	0.244	Phosphate acetyltransferase
Benzoate degradation	*VP_RS20635*	0.295	Carboxymuconolactone decarboxylase family protein
	*VP_RS20550*	2.679	Thiolase family protein
	*VP_RS00135*	2.713	Fatty acid oxidation complex subunit alpha FadB
RNA transport	*VP_RS19430*	0.440	Stress response translation initiation inhibitor YciH
	*VP_RS01980*	0.485	Multifunctional CCA addition/repair protein
Phosphonate and phosphinate metabolism	*VP_RS16410*	0.206	2-aminoethylphosphonate--pyruvate Transaminase
	*VP_RS16400*	0.491	Phosphonoacetaldehyde hydrolase
Ethylbenzene degradation	*VP_RS10720*	2.111	Acetyl-CoA C-acyltransferase FadI
	*VP_RS00130*	2.465	Acetyl-CoA C-acyltransferase FadA
Biotin metabolism	*VP_RS05435*	0.057	Dethiobiotin synthase
	*VP_RS21415*	0.265	Beta-ketoacyl-ACP reductase
	*VP_RS05415*	0.376	Adenosylmethionine-8-amino-7-oxononanoate transaminase
	*VP_RS05425*	0.454	8-amino-7-oxononanoate synthase
	*VP_RS05420*	0.479	Biotin synthase BioB
	*VP_RS05430*	0.492	Malonyl-ACP O-methyltransferase BioC
	*VP_RS20520*	2.061	SDR family oxidoreductase

**Table 6 molecules-27-00660-t006:** Major altered metabolic pathways in *V. parahaemolyticus* B4-10 treated by the CC1 from *R. madaio* Makino.

Metabolic Pathway	Gene ID	Fold Change	Gene Description
Styrene degradation	*VP_RS06550*	0.394	Homogentisate 1%2C2-dioxygenase
	*VP_RS06560*	0.408	Maleylacetoacetate isomerase
	*VP_RS06555*	0.471	Fumarylacetoacetate hydrolase family protein
Nitrogen metabolism	*VP_RS20240*	2.129	Nitrite reductase large subunit NirB
	*VP_RS19890*	2.518	Nitrite reductase small subunit NirD
	*VP_RS20235*	2.823	Nitrite reductase small subunit NirD
	*VP_RS20280*	3.753	Nitrate reductase
	*VP_RS20915*	3.759	Nitrate reductase cytochrome c-type subunit
	*VP_RS19895*	3.988	Nitrite reductase large subunit NirB
	*VP_RS20910*	4.186	Periplasmic nitrate reductase subunit alpha
	*VP_RS20250*	10.250	ABC transporter permease
	*VP_RS09370*	29.586	Ammonia-forming nitrite reductase cytochrome c552 subunit
	*VP_RS05780*	107.754	Hydroxylamine reductase
Quorum sensing	*VP_RS06530*	0.241	Oligopeptide ABC transporter permease OppB
	*VP_RS06520*	0.256	ATP-binding cassette domain-containing protein
	*VP_RS06525*	0.265	ABC transporter permease subunit
	*VP_RS06515*	0.297	ATP-binding cassette domain-containing protein
	*VP_RS06485*	0.310	ABC transporter ATP-binding protein
	*VP_RS06495*	0.346	ABC transporter permease
	*VP_RS06535*	0.362	Peptide ABC transporter substrate-binding protein
	*VP_RS20670*	0.368	ABC transporter ATP-binding protein
	*VP_RS06490*	0.370	ABC transporter permease
	*VP_RS20680*	0.381	Branched-chain amino acid ABC transporter permease
	*VP_RS06470*	0.388	Polyamine ABC transporter substrate-binding protein
	*VP_RS21025*	0.416	Autoinducer 2-binding periplasmic protein LuxP
	*VP_RS20695*	0.455	ABC transporter ATP-binding protein
	*VP_RS01695*	0.468	Long-chain fatty acid--CoA ligase
	*VP_RS20675*	0.475	ABC transporter substrate-binding protein
	*VP_RS00850*	0.495	ABC transporter ATP-binding protein
	*VP_RS12050*	2.098	ABC transporter ATP-binding protein
	*VP_RS15305*	2.117	GTP cyclohydrolase II
	*VP_RS22315*	2.159	ABC transporter ATP-binding protein
	*VP_RS12040*	2.232	ABC transporter permease
	*VP_RS08360*	2.551	Two-component sensor histidine kinase
	*VP_RS22015*	2.976	Response regulator transcription factor
	*VP_RS08355*	3.014	Response regulator
	*VP_RS16930*	3.141	Permease
Folate biosynthesis	*VP_RS17975*	0.476	Phenylalanine 4-monooxygenase
	*VP_RS09130*	0.494	Aminodeoxychorismate synthase component I
	*VP_RS03365*	0.491	NADPH-dependent 7-cyano-7-deazaguanine reductase QueF
	*VP_RS07885*	0.497	7-cyano-7-deazaguanine synthase QueC
	*VP_RS09170*	0.389	6-carboxytetrahydropterin synthase QueD
	*VP_RS13730*	0.433	Aminodeoxychorismate/anthranilate synthase component II
	*VP_RS07890*	0.484	7-carboxy-7-deazaguanine synthase QueE
	*VP_RS17980*	0.432	4a-hydroxytetrahydrobiopterin dehydratase
	*VP_RS01970*	0.431	2-amino-4-hydroxy-6-hydroxymethyldihydropteridine diphosphokinase
Histidine metabolism	*VP_RS06185*	10.231	Urocanate hydratase
	*VP_RS06180*	6.284	Histidine ammonia-lyase
	*VP_RS06195*	6.998	Imidazolonepropionase
	*VP_RS06190*	5.106	Formimidoylglutamase
	*VP_RS05565*	0.496	Bifunctional phosphoribosyl-AMP cyclohydrolase/phosphoribosyl-ATP diphosphatase HisIE

**Table 7 molecules-27-00660-t007:** Major altered metabolic pathways in *B. cereus* A1-1 treated by the CC1 from *R. madaio* Makino.

Metabolic Pathway	Gene ID	Fold Change	Gene Description
Flagellar assembly	*BCN_RS08555*	0.038	Flagellar assembly protein FliH
	*BCN_RS08605*	0.045	Flagellin
	*BCN_RS08610*	0.072	Flagellin
	*BCN_RS08640*	0.108	Flagellar type III secretion system pore protein FliP
	*BCN_RS08550*	0.113	Flagellar motor switch protein FliG
	*BCN_RS22265*	0.115	Flagellar motor stator protein MotA
	*BCN_RS22260*	0.143	Flagellar motor protein MotB
	*BCN_RS08545*	0.154	Flagellar M-ring protein FliF
	*BCN_RS08470*	0.158	Flagellar motor switch protein
	*BCN_RS08560*	0.158	Flagellar protein export ATPase FliI
	*BCN_RS08535*	0.173	Flagellar basal body rod protein FlgC
	*BCN_RS08670*	0.188	Flagellar basal-body rod protein FlgG
	*BCN_RS08520*	0.196	Flagellar protein FliS
	*BCN_RS08530*	0.197	Flagellar basal body rod protein FlgB
	*BCN_RS08625*	0.200	Flagellar motor switch protein FliM
	*BCN_RS08660*	0.230	Flagellar biosynthesis protein FlhA
	*BCN_RS08510*	0.241	Flagellar hook-associated protein 3
	*BCN_RS08655*	0.392	Flagellar type III secretion system protein FlhB
	*BCN_RS08650*	0.438	Flagellar type III secretion system protein FliR
Bacterial chemotaxis	*BCN_RS10010*	0.063	Methyl-accepting chemotaxis protein
	*BCN_RS03675*	0.088	Methyl-accepting chemotaxis protein
	*BCN_RS02280*	0.185	Methyl-accepting chemotaxis protein
	*BCN_RS08460*	0.186	Response regulator
	*BCN_RS08625*	0.200	Flagellar motor switch protein FliM
	*BCN_RS25160*	0.265	DUF4077 domain-containing protein
	*BCN_RS24975*	0.321	Methyl-accepting chemotaxis protein
	*BCN_RS08595*	0.357	Chemotaxis protein
	*BCN_RS08455*	0.474	OmpA family protein
Two-component system	*BCN_RS27005*	0.136	Respiratory nitrate reductase subunit gamma
	*BCN_RS26190*	0.152	Cytochrome d ubiquinol oxidase subunit II
	*BCN_RS23710*	0.219	Potassium-transporting ATPase subunit KdpA
	*BCN_RS27000*	0.231	Acetyl-CoA C-acyltransferase
	*BCN_RS23715*	0.258	Methyl-accepting chemotaxis protein
	*BCN_RS04080*	0.385	Nitrate reductase molybdenum cofactor assembly chaperone
	*BCN_RS15080*	0.401	Response regulator
	*BCN_RS04090*	0.419	Methyl-accepting chemotaxis protein
	*BCN_RS07505*	2.006	Phosphate ABC transporter substrate-binding protein PstS
	*BCN_RS26540*	2.297	Cytochrome ubiquinol oxidase subunit I
	*BCN_RS17290*	2.348	Chemotaxis protein CheA
	*BCN_RS02700*	3.703	Antiholin-like murein hydrolase modulator LrgA
	*BCN_RS10795*	4.600	Acetyl-CoA C-acetyltransferase
	*BCN_RS07495*	5.804	Hypothetical protein
Thiamine metabolism	*BCN_RS29465*	0.031	TenA family transcriptional regulator
	*BCN_RS02365*	0.205	Thiamine phosphate synthase
	*BCN_RS04005*	0.224	Thiaminase II
	*BCN_RS04040*	0.274	Thiazole synthase
	*BCN_RS04030*	0.282	Glycine oxidase ThiO
	*BCN_RS04050*	0.304	Bifunctional hydroxymethylpyrimidine kinase/phosphomethylpyrimidine kinase
	*BCN_RS04025*	0.310	Thiazole tautomerase TenI
	*BCN_RS25935*	0.320	Phosphomethylpyrimidine synthase ThiC
	*BCN_RS21485*	0.342	Alkaline phosphatase
	*BCN_RS12695*	0.397	Thiaminase II
	*BCN_RS02360*	0.407	Hydroxyethylthiazole kinase
	*BCN_RS10005*	0.407	Ribosome small subunit-dependent GTPase A
	*BCN_RS22955*	0.433	Cysteine desulfurase
	*BCN_RS02660*	0.457	Acetylornithine deacetylase
ABC transporters	*BCN_RS03130*	0.051	Amino acid ABC transporter permease
	*BCN_RS14125*	0.051	Glycine betaine ABC transporter substrate-binding protein
	*BCN_RS15895*	0.056	Substrate-binding domain-containing protein
	*BCN_RS06920*	0.179	ABC transporter ATP-binding protein
	*BCN_RS17880*	0.205	Ribose ABC transporter ATP-binding protein RbsA
	*BCN_RS01110*	0.221	Amino acid ABC transporter ATP-binding protein
	*BCN_RS06915*	0.225	Peptide ABC transporter substrate-binding protein
	*BCN_RS01100*	0.258	Amino acid ABC transporter ATP-binding protein
	*BCN_RS04010*	0.263	Phosphate ABC transporter permease PstA
	*BCN_RS08770*	0.268	Peptide ABC transporter substrate-binding protein
	*BCN_RS14120*	0.268	BMP family protein
	*BCN_RS20515*	0.272	ABC transporter ATP-binding protein
	*BCN_RS03855*	0.278	Phosphonate ABC transporter ATP-binding protein
	*BCN_RS01165*	0.282	Molybdate ABC transporter permease subunit
	*BCN_RS20525*	0.283	ABC transporter ATP-binding protein
	*BCN_RS21100*	0.320	Metal ABC transporter substrate-binding protein
	*BCN_RS04020*	0.322	ABC transporter substrate-binding protein
	*BCN_RS04015*	0.326	Phosphate ABC transporter permease subunit PstC
	*BCN_RS03845*	0.330	ATP-binding cassette domain-containing protein
	*BCN_RS03850*	0.347	Phosphate ABC transporter ATP-binding protein
	*BCN_RS24655*	0.347	Transporter substrate-binding domain-containing protein
	*BCN_RS01125*	0.351	Putative 2-aminoethylphosphonate ABC transporter ATP-binding protein
	*BCN_RS20520*	0.355	Aliphatic sulfonate ABC transporter substrate-binding protein
	*BCN_RS18335*	0.379	Iron ABC transporter permease
	*BCN_RS09350*	0.405	Energy-coupling factor transporter transmembrane protein EcfT
	*BCN_RS24665*	0.405	Putative 2-aminoethylphosphonate ABC transporter substrate-binding protein
	*BCN_RS01160*	0.413	Molybdate ABC transporter substrate-binding protein
	*BCN_RS04750*	0.458	ABC transporter permease
	*BCN_RS01870*	0.465	ABC transporter permease
	*BCN_RS17755*	0.470	Methionine ABC transporter substrate-binding lipoprotein MetQ
	*BCN_RS03600*	0.487	Phosphate ABC transporter substrate-binding protein PstS
	*BCN_RS09570*	0.487	Peptide ABC transporter substrate-binding protein
	*BCN_RS10085*	0.487	Sugar ABC transporter permease
	*BCN_RS09640*	4.508	Thiol reductant ABC exporter subunit CydC
	*BCN_RS26090*	14.65	ABC transporter substrate-binding protein
	*BCN_RS13495*	20.285	MetQ/NlpA family ABC transporter substrate-binding protein
Arginine biosynthesis	*BCN_RS20420*	0.070	N-acetyl-gamma-glutamyl-phosphate reductase
	*BCN_RS20400*	0.117	Ornithine carbamoyltransferase
	*BCN_RS20410*	0.159	Acetylglutamate kinase
	*BCN_RS20405*	0.171	Acetylornithine transaminase
	*BCN_RS20415*	0.271	Bifunctional glutamate N-acetyltransferase/amino-acid acetyltransferase ArgJ
	*BCN_RS00945*	0.281	Arginase
	*BCN_RS22860*	0.292	Argininosuccinate lyase
	*BCN_RS22865*	0.486	Argininosuccinate synthase
Nitrogen metabolism	*BCN_RS07150*	0.365	Nitronate monooxygenase
	*BCN_RS10835*	5.001	Nitrate transporter NarK
	*BCN_RS10790*	6.281	Nitrate reductase subunit beta
	*BCN_RS10800*	7.880	Respiratory nitrate reductase subunit gamma
	*BCN_RS10785*	8.675	Nitrate reductase subunit alpha
	*BCN_RS10870*	8.912	Nitrite reductase small subunit NirD
	*BCN_RS10875*	15.156	NADPH-nitrite reductase large subunit
	*BCN_RS16540*	150.780	Hydroxylamine reductase
Riboflavin metabolism	*BCN_RS20310*	3.325	Bifunctional diaminohydroxyphosphoribosylaminopyrimidine deaminase/5-Amino-6-(5-phosphoribosylamino) uracil reductase RibD
	*BCN_RS20320*	4.247	Bifunctional 3%2C4-dihydroxy-2-butanone 4-phosphate synthase/GTP Cyclohydrolase II
	*BCN_RS20325*	4.361	6%2C7-dimethyl-8-ribityllumazine synthase
	*BCN_RS20315*	4.769	Riboflavin synthase subunit alpha
Pyrimidine metabolism	*BCN_RS15125*	0.304	5’-nucleotidase C-terminal domain-containing protein
	*BCN_RS24625*	0.355	Bifunctional metallophosphatase/5’-nucleotidase
	*BCN_RS18815*	0.381	Carbamoyl-phosphate synthase large subunit
	*BCN_RS18820*	0.406	Carbamoyl phosphate synthase small subunit
	*BCN_RS18795*	0.419	Orotate phosphoribosyltransferase
	*BCN_RS18805*	0.430	Dihydroorotate oxidase B catalytic subunit
	*BCN_RS18800*	0.438	Orotidine-5’-phosphate decarboxylase
	*BCN_RS18810*	0.441	Dihydroorotate oxidase B electron transfer subunit
	*BCN_RS20265*	0.445	5’-nucleotidase C-terminal domain-containing protein
	*BCN_RS18825*	0.449	Dihydroorotase
	*BCN_RS07895*	0.462	Nucleoside-diphosphate kinase
	*BCN_RS09440*	0.473	Pyrimidine-nucleoside phosphorylase
HIF-1 signaling pathway	*BCN_RS24725*	0.191	L-lactate dehydrogenase
	*BCN_RS25405*	2.598	Phosphoglycerate kinase
	*BCN_RS25410*	2.736	Type I glyceraldehyde-3-phosphate dehydrogenase
	*BCN_RS25390*	3.143	phosphopyruvate hydratase
	*BCN_RS24095*	5.531	L-lactate dehydrogenase
Fatty acid degradation	*BCN_RS17445*	0.340	Acetyl-CoA C-acetyltransferase
	*BCN_RS17450*	0.456	Acyl-CoA synthetase
Alanine, aspartate and glutamate metabolism	*BCN_RS08845*	0.353	Glutaminase A
	*BCN_RS08855*	0.361	Hypothetical protein
	*BCN_RS19905*	0.420	Carbon-nitrogen family hydrolase
	*BCN_RS15030*	0.486	Asparaginase
	*BCN_RS03305*	0.498	Aspartate ammonia-lyase
	*BCN_RS00970*	2.986	Glutamine--fructose-6-phosphate transaminase (isomerizing)
	*BCN_RS03230*	7.200	Alanine dehydrogenase
Benzoate degradation	*BCN_RS26535*	2.191	3-hydroxybutyryl-CoA dehydrogenase
	*BCN_RS24780*	2.199	Acetyl-CoA C-acetyltransferase
	*BCN_RS24785*	2.285	3-hydroxyacyl-CoA dehydrogenase/enoyl-CoA hydratase family protein
Glycolysis/Gluconeogenesis	*BCN_RS08815*	0.225	Histidine phosphatase family protein
	*BCN_RS21600*	0.299	Bifunctional acetaldehyde-CoA/alcohol dehydrogenase
	*BCN_RS11285*	0.411	Alcohol dehydrogenase AdhP
	*BCN_RS28275*	0.413	S-(hydroxymethyl)glutathione dehydrogenase/class III alcohol dehydrogenase
	*BCN_RS22940*	0.489	Acyl-CoA ligase
	*BCN_RS26420*	2.666	PTS glucose transporter subunit IIA
	*BCN_RS25395*	2.901	2%2C3-bisphosphoglycerate-independent phosphoglycerate mutase
	*BCN_RS25815*	5.561	6-phospho-beta-glucosidase
Inositol phosphate metabolism	*BCN_RS18155*	0.186	Phosphatidylinositol diacylglycerol-lyase
	*BCN_RS03640*	0.245	Phospholipase C
	*BCN_RS25400*	2.616	Triose-phosphate isomerase
Butanoate metabolism	*BCN_RS02750*	0.158	Formate C-acetyltransferase
	*BCN_RS07305*	0.199	Acetolactate synthase large subunit
	*BCN_RS11410*	0.359	Acetate CoA-transferase subunit alpha
	*BCN_RS11415*	0.382	CoA transferase subunit B
	*BCN_RS04800*	2.474	Alpha-acetolactate decarboxylase
Propanoate metabolism	*BCN_RS18555*	0.407	ADP-forming succinate--CoA ligase subunit beta
	*BCN_RS07995*	0.451	Methylglyoxal synthase
	*BCN_RS18550*	0.467	Succinate-CoA ligase subunit alpha

## Data Availability

A complete list of DEGs in the four strains were available in the NCBI SRA database (https://submit.ncbi.nlm.nih.gov/subs/bioproject/, accessed on 17 October 2021) under the accession number PRJNA767551.

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
