# Peer review of "Identification of Antibacterial Components in the Methanol-Phase Extract from Edible Herbaceous Plant Rumex madaio Makino and Their Antibacterial Action Modes"

_molecules, 2022, doi:10.3390/molecules27030660_

Round 1
Reviewer 1 Report
Comments and suggestions to authors
- The title is confusing, please rephrase it.
- The manuscript lack of discussion part, please provide this section in the manuscript.
- The UHPLC-MS analysis in table 3 for the fraction CC1 revealed that most of the identified compounds are of saccharides, amino acids, and simple alkaloids which mostly in active as antimicrobial agent. Please justify this in the discussion part; you may also need to change the parameter of the UHPLC-MS analysis to show more secondary metabolites in the extract.
- Introduction: I think that paragraphs 1 and 2 of no use and should be deleted.
- Results:
Figure 1: please provide the negative control for each strain.
Table 3: compound one is phenol of alkaloid?
Compound two is amino acid
Compound 11 and 12,what authors mean by “Carboxylic acids and derivatives”, same in compound 17 and 18.
Compound 24 is not fatty acyls,
Please check the identity and nature of all other compounds in table 3
- Methods: Please provide a detailed methodology for the UHPLC- MS analysis
29.11.2021
Reviewer 2 Report
Authors have applied chromatography and mass spectrometry to purify and identify antibacterial compounds from the edible plant R. madaio, and used transcriptomic analysis to show that these compounds can alter gene expression that are central to bacterial survival and pathogenicity.
I find the manuscript to be scientifically sound and well-written. I do not have any major concern provided the authors clarify a few points by adding some comments in the manuscript.
Minor points:
- What was the basis for selecting the 23 species of pathogenic bacteria as shown in Table1? For eg- Acinetobacter baumannii, Mycobacterium tuberculosis, etc also deserve attention while discussing the global concern on common pathogenic bacteria. If these bacterial samples (along with many others) were indeed considered, then authors must include comments regardless whether a positive or negative results were obtained.
- Minimum inhibitory concentration (MIC) should be spelled out at the first instance in page 3 (see Table1).
- Materials and methods 3.2: please mention the probe size used for sonication.
Round 2
Reviewer 1 Report
Dear authors,
Thank you for reply and implementation of the recommendation, manuscript now is fine and I recommend its publication in the current form.
You could be just delete the words "and derivatives" from table three.
Regards